# Causes and Treatment Measures of Submarine Pipeline Free-Spanning

**Bo Zhang [1], Rui Gong [1], Tao Wang [2] and Zhuo Wang [1,\*]**

[1]  College of Mechanical and Electrical Engineering, Harbin Engineering University, Harbin 150001, China; wangzhuo_heu@hrbeu.edu.cn (B.Z.); shangzhiquan@hrbeu.edu.cn (R.G.)
[2]  School of MechanicalEngineering, Hebei University of Technology, Tianjin 300401, China; 18846166436@hrbeu.edu.cn
\*  Correspondence: zhangbo_heu@hrbeu.edu.cn

**Abstract:** Submarine pipelines, as arteries for offshore oil and gas transportation, play a particularly important role in the exploitation of offshore oil and gas resources. Since the world's first submarine pipelines were laid in the Gulf of Mexico, numerous failures have been caused by pipeline free-spanning. This paper provides a review of the causes and treatment measures for the free span of submarine pipeline. Various factors cause the free span of submarine pipelines, including wave flow scouring, fluctuations in seabed topography, residual stress or thermal stress of pipelines, and human activities. The scour of the wave current is the main factor affecting free span; the research on sediment starting and equilibrium depth during scour is reviewed in-depth. For the span treatment of submarine pipelines, the main measures available at present include the re-digging trench burying, structural support, covering bionic water plants, and choke plate self-burying. For each, the principle, advantages, disadvantages, and research are discussed. This review provides a convenient resource for understanding the causes of submarine free-spanning pipelines and choosing suitable treatment measures.

**Keywords:** oil and gas transportation; submarine pipelines; free-spanning; causes; treatment measures; scour

## 1. Introduction

Energy is the material basis for the development of human society. All countries in the world have experienced a stage of promoting the development of society through the extensive use of fossil energy. During this period, human society has developed tremendously, and human civilization, in the fields of natural sciences and humanities, has reached unprecedented heights [1]. However, due to excessive pursuit of development speed, human exploitation and use of fossil fuels have aggravated the global warming effect of the earth and caused serious air pollution. With the excessive exploitation of fossil energy, human beings are facing an unprecedented energy crisis [2]. Therefore, the development of the use of new energy to alleviate the pressure on fossil energy is urgently required.

With the improvement of exploration technology, a large number of oil and gas resources buried in the ocean have been discovered [3]. Since Brown and Root laid the world's first submarine pipelines in the Gulf of Mexico in 1954, the total length of submarine pipelines laid in countries around the world has reached more than 100,000 km [4].

However, due to the harsh and complex marine environment, the application of submarine pipelines faces enormous challenges. The pipelines laid on the seabed often have overhangs due to the ups and downs of the seabed topography, the movement of the seabed, the scour of the seabed by currents and waves, the abnormal tidal currents and storms, and the residual stress and thermal

stress of the pipelines [5]. Submarine pipelines with major free spans, if not treated in time, may lead to fatigue damage or strength damage to the pipelines, causing oil and gas leakage, resulting in huge economic losses and major marine pollution accidents. Submarine pipeline free-spanning has become one of the major risks and hazards in the development of marine oil and gas resources [6]. Therefore, studying the formation mechanism and treatment measures of free-spanning pipelines is essential [7].

In this paper, the causes of submarine pipelines free-spanning are first summarized. The current research status of wave scour, which is the most common cause of submarine pipeline free-spanning, is introduced. Then, the various methods in the governance of pipeline free-spanning are introduced.

## 2. Causes of Submarine Pipeline Free-Spanning

Free-spanning refers to the suspended pipeline section where there is no direct contact between the submarine pipelines and the seabed surface for some reason [8]. Many factors cause the free-spanning of submarine pipelines. The main reasons include the scour of ocean current, uneven seabed topography, human activities, and residual or thermal stress. Table 1 summarizes the failure accidents caused by the free-spanning of submarine pipelines that have occurred throughout history.

**Table 1.** Failure accidents caused by spanning of submarine pipelines.

| Year | Place | Reason | Consequence |
|------|-------|--------|-------------|
| 1964–1965 | U.S. offshore water pipelines | Impact of Hurricane Betsy | Pipelines moved |
| 1996 | U.S. offshore water pipelines | Impact of Hurricane Flossy | Non-buried pipelines moves |
| 1997 | The Zeepipe IIA pipelines in the Norwegian sector of the North Sea | Wave scouring | Exceeding the maximum span, threatening security operations |
| 2001 | Daixi landing pipelines | Scouring leads to insufficient depth | Ship drag anchor damage |
| 2002 | Yubei Oilfield | Wave scouring | Pipe fatigue fracture |
| 2003–2009 | Chengdao Oilfield | Wave scouring | Pipe rupture, causing oil and gas leaks |
| 2009 | Ledong gas field | Pipelines climbing and scouring | Exceeding the maximum span, threatening security operations |
| 2010 | Huizhou Natural Gas Submarine Pipelines | Scouring | Pipe fatigue cracking |

Table 1 shows that the submarine pipeline free-spanning caused by wave scouring is the most important cause of failure of submarine oil and gas pipelines. Therefore, reviewing the mechanisms and research status of pipelines scouring is necessary.

A trend in the offshore oil and gas pipelines industry is the planning and installation of submarine pipelines in areas where the terrain of the deep ocean floor is increasingly irregular. The uneven topography of the sea floor is an important reason for the overhang of the pipelines. This type of pipeline includes new pipelines that pass through uneven seabeds in the Mediterranean and North Sea and deep-water pipelines that cross the uneven continental slope of the Gulf of Mexico [9], the gas pipelines from Tunisia to Italy, the Maghreb gas pipelines from Algeria to Spain across the Strait of Gibraltar, the Troll and Haltenpipe gas pipelines from the North Sea to Norway, and the Fensfjorden on the Norwegian coast. Pipelines constructed in these areas experience a series of continuous free spans, and pipelines engineers face significant challenges in establishing safe and economical design and installation solutions [10,11].

### 2.1. Free-Spanning of Sea Pipelines Caused by Wave Current Scouring

Submarine-laid pipelines introduce local disturbances in wave flow and offshore flow. When flow affects the pipelines, local velocity fields and pressure gradients are created between the upstream

and downstream of the pipelines. If the sea floor is permeable, that is, sandy, this pressure gradient promotes seepage under the pipelines. This seepage tends to drag sand particles. If there is enough energy, the sand under the pipelines can be washed away. This is the beginning of scour under the pipelines. Scouring occurs when the pressure gradient on both sides of the pipe exceeds the stabilizing forces that keep the sediment in place: gravity, particle friction, and the submerged weight of the covered pipe. The initial scour may be local, including a length that is several times shorter than the diameter of the pipe, or it may be global. The former case is typically scouring caused by a steady stream of water, and the latter is classified as tunnel erosion [12].

Arnold [13] and Demars [14] analyzed submarine pipelines accidents of the Mississippi River delta and the Gulf of Mexico in the United States, respectively. They found that pipeline free-spanning caused by scour is one of the main causes of submarine pipeline failure. Through theoretical analysis, model experiment, and numerical simulation, scholars around the world have extensively researched the sediment initiation and equilibrium depth of the scour of submarine pipelines.

In terms of scour initiation, two main methods are used to evaluate whether the seabed at the bottom of the submarine pipelines will scour. One is based on the pressure gradient analysis on both sides of the counter current and back flow of the seabed at the bottom of the pipelines, and the other is based on the critical velocity point of the water quality at which the movement of sand particles starts [15].

Zang et al. [16] performed equilibrium stress analysis on sediment particles under the action of constant flow and wave alone. The sediment initiation formula was obtained. The critical condition of sediment initiation under the action of constant flow is shown in Equation (1):

$$\left[\frac{v^2}{gD(1-n)(s-1)}\right]_{C_r} \geq \frac{\gamma}{\lambda_A \Delta C_p} \tag{1}$$

and the critical condition formula of sediment starting under wave action is shown in Equation (2).

$$\frac{v^2}{gD(1-n)(s-1)} \geq \frac{\gamma}{\lambda_A \Delta C_p}\left[1 - \exp\left(-0.018K^{1.5}\right)\right] \tag{2}$$

here $g$ is the acceleration due to gravity, $D$ is the outer diameter of the pipe, $n$ is the porosity of the sand, $s$ is the relative specific gravity of the sand with respect to water, $v$ is the critical velocity, $\Delta C_p$ is the pressure difference coefficient before and after the pipelines, $\lambda_A$ is the correlation coefficient; $\gamma$ is the angle between the pipeline axis and the sea floor, and $K$ is in the Keulegan–Carpenter number, which is defined as:

$$Kc = \frac{U_m T}{D} \tag{3}$$

here $U_m$ is the maximum horizontal velocity of the wave water point near the bottom of the tube, $T$ is the wave period.

Sediment initiation is the first step in the whole scour process. Sumer [17] divided the piping phenomenon in the initial stage of pipelines scouring into two stages by studying the details of scouring. In the first stage, when the incoming stream acts for a period of time, the seepage force of water in the seabed begins to be greater than the floating weight of particles, and the surface of sand begins to uplift. In the second stage, piping occurs, that is, a mixture of water and sand will gush out the piping. The piping process is shown in Figure 1a–c. Previous studies on sediment initiation generally simplified the seabed surface as a rigid impermeable boundary, ignoring the influence of seepage on the mobility of bed particles. To reveal the physical law of sediment initiation in the early stage of scour of free-spanning pipelines, Xia [18] analyzed the influence of seepage force generated by wave seepage in the boundary layer on sediment starting. Zhen [19] proposed a numerical simulation method that couples the shear stress sediment transport turbulence model with the porous seabed model. Studies showed that as the vortex around the pipe periodically forms and falls off, the bottom

of the sea generates oscillations and residual excess pore pressure. In some cases, the vertical gradient of the superpose pressure (seepage force) significantly affects the fluidity of the bed particles around the pipe. Mattioli et al. used numerical calculations to study the turbulence and vortices of the local scour of submarine pipelines laid on a non-cohesive sandy sea floor under the action of stable water flow. The numerical model uses the level set technique to solve the Navier–Stokes equation. The model predicts the behavior of moving sediment through two components of drift force and lift force [20].

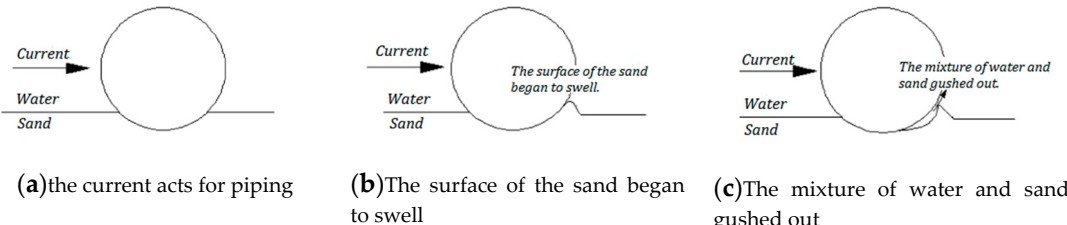

(**a**)the current acts for piping   (**b**)The surface of the sand began to swell   (**c**)The mixture of water and sand gushed out

**Figure 1.** Sediment initiation during the initial scour phase. (**a**) The current acts for piping, (**b**) The surface of the sand began to swell, (**c**) The mixture of water and sand gushed out.

Research on the scour balance depth started in the 1960s. From the late 1980s until the 1990s, Sumer et al. [21–26] and Chiew et al. [27–30] extensively examined the scour of the seabed around submarine pipelines. Myrhaug et al. [31] studied the formula for the depth and width of the scour pit under the action of random waves and the depth of the scour pit around the pile. Kumar et al. [32] analyzed the relationship between the maximum scour depth of submarine pipelines on the viscous sea floor and the consistency of clay using wave tank experiments. Yan et al. [33] analyzed the scour stability of submarine pipelines when buried in a certain depth of soil and exposed on the seabed surface. The maximum scour depth at scour equilibrium was calculated under two conditions. Feng [34] used a large-scale flume experiment to study the scouring of silt around the pile under the action of waves and flow alone, and waves and flow together. A formula for calculating the maximum scour depth was derived. According to the different water flow conditions, the scour problem can be divided into two types: clean water scour ($\theta < \theta_{cr}$) and live bed scour ($\theta > \theta_{cr}$), where $\theta$ is the Shields parameter of the sediment particles and $\theta_{cr}$ is the critical Shields parameter [35–37]. The scouring mechanism under clear water conditions is different from that under live bed conditions. Dey et al. [38] and Moncada et al. [39] conducted pipe-scouring experiments under clean water and live bed conditions, respectively. Dey et al. [40] established a semi-theoretical calculation model for the maximum depth of clean water scour under a pipeline, and studied the effect of upward seepage on the depth of clean water scour under the pipeline through experiments. Ibrahim and Nalluri [41] analyzed experimental results and reported that in the case of clean water scouring, no scouring away from the pipeline occurs; in the case of moving bed scouring, the entire bottom bed will scour. This laid the foundation for future research on scouring issues.

For submarine pipelines laid in shallow coastal waters, the effect of waves on local erosion of submarine pipelines cannot be ignored. The two-dimensional pipeline scour problem under the action of waves is different from the pipeline scour problem under the action of currents. This is mainly reflected in vortex shedding occurring in front of and behind the pipeline under the action of waves. This causes the scour to develop back and forth toward the pipeline. The *KC* number is an important parameter representing the range of motion of water particles. A large *KC* indicates that the water particle has a large range of motion, which causes a large range of local scour around the pipeline. A small KC number indicates that the water particle has a small range of motion, which causes a small range of pipeline erosion. Many experiments have been conducted on wave-induced scour under

pipelines [42–44]. Sume et al. [45] found through experiments that the local maximum depth of the pipeline and the KC number have the following relationship:

$$\frac{S}{D} = 0.1 \sqrt{KC} \qquad (4)$$

here *S* is the scour depth and *D* is the pipe diameter. This shows that the *KC* number is the main factor affecting the local scour depth of subsea pipelines under the action of waves.

Studies on the scouring of submarine pipelines have mainly linked the formation of scouring to the pressure gradient on both sides of the pipeline. Mattioli et al. [46,47] used particle tracking technology (PTV) to study the motion and evolution of particles caused by water waves around a cylinder on a rigid or erodible ocean floor. They also analyzed the flow field around the pipeline under different KC numbers, and reported that the vortex structure around the pipeline may be combined with the pressure gradient and play an important role in the process of erosion.

Chang et al. [48] used the Navier–Stokes equation and Suagorinsky sub-grid model to simulate the development and change in scour in the process of pipelines subsidence. They found that the depth of scour under the pipelines gradually increased with the decrease in pipelines subsidence velocity. When the subsidence velocity was sufficient, the scour pit depth could reach more than 1*D*. Li and Cheng [49] proposed a mathematical model based on potential flow theory to simulate the scour of submarine pipelines under the action of unidirectional flow. After verification, this model can be used to predict the scour depth under the condition of clear water scour.

Most of the research on scouring of submarine pipelines has focused on sandy seabeds, whereas research on scouring of pipelines on cohesive soils is lacking. No complete formula exists for predicting the scour depth of pipes on the viscous sea floor. In 2015, Postacchini and Brocchini analyzed the influence of the amount of clay in the seabed on the depth of pipeline erosion by collating the collected data. For the first time, they tried to establish a prediction formula for the depth of pipeline erosion when the clay content is low, laying the foundation for future research on the scouring depth of pipelines on the viscous sea floor [50].

Many scholars have conducted many experimental studies and numerical calculations to predict erosion under pipelines [51–54]. In addition to traditional methods, artificial intelligence methods have been applied to pipeline erosion prediction, such as artificial neural networks (ANNs), machine learning methods, adaptive neural fuzzy inference systems (ANFIS), and genetic programming (GP) [55–58].

Many scholars have summarized the relevant research results and work experience with pipeline overhang. Bijker et al. [59] reviewed the origin and role of scour-induced free-crossing as the self-descending process of submarine pipelines. The single-span development is described and discussed from the aspects of initial embedding, scouring start, scouring development rate, relevant stopping standards, multi-span formation, and critical free-span development. Sumer et al. [60] reviewed the research progress of pipe perimeter scour under non-cohesive sediment. The review was divided into four parts: two-dimensional scouring, three-dimensional scouring, the influence of scouring on the force and vibration of the pipeline, and a mathematical model of the scouring process. More than 60 works were included in the review. Drago et al. [61] discussed more than 30 years of research and development achievements and project experience, summarized the underwater environment and current conditions near the overhang pipeline, and discussed the occurrence and development of overhang. In addition, they summarized the design method of free-spanning pipelines and some pipelines inspection experiences.

The free-spanning of submarine pipelines caused by ocean current is one of the main causes of submarine oil and gas pipelines failure. Therefore, many scholars have completed large amounts of research on this issue. Factors affecting pipelines scour include the initial buried depth of the pipelines, particle size of sediment, diameter of the pipelines, KC number, clay content, wave and flow effects, etc. [62–67]. Because influencing factors are numerous and complex, current research is generally aimed at one or a specific number of factors which is a certain deviation from actual engineering.

### 2.2. Free-Spanning of Sea Pipe Caused by Other Reasons

### 2.2.1. Free-Spanning of Sea Pipe Caused by Seabed Relief

In the process of laying submarine pipelines, some regions will have considerable changes in submarine topography. This type of free span shows different spanning forms due to the differences in topography and geology of the route area [68,69], as shown in Figure 2.

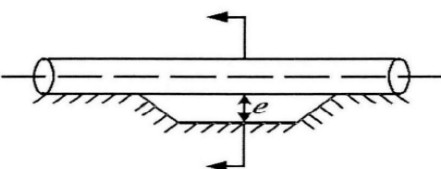

(**a**) Overhangs formed by uneven seabed

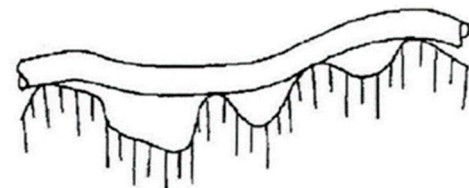

(**b**) Sectional overhangs caused by topographic relief

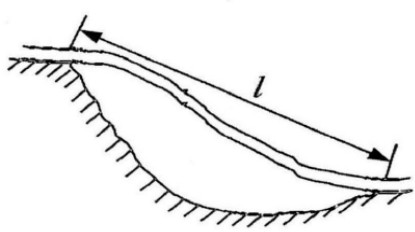

(**c**) Free-spanning formed by pipeline climbing

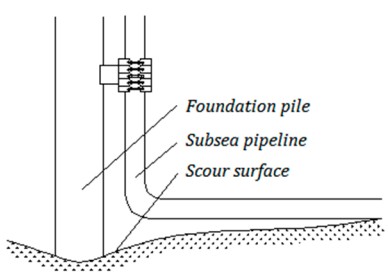

(**d**) Free-spanning formed by connection with offshore platform riser

**Figure 2.** Different free-spanning forms caused by seabed topography differences. (**a**) Overhangs formed by uneven seabed, (**b**) Sectional overhangs caused by topographic relief, (**c**) Free-spanning formed by pipeline climbing, (**d**) Free-spanning formed by connection with offshore platform riser.

Wen et al. [70] measured the abandoned submarine pipelines in the Yellow River delta and found that the topography of the seabed strongly influenced the span of the pipelines. Therefore, the choice of the laying route of the submarine pipelines is crucial. Finding a completely flat laying route is usually impossible. However, a suitable pipeline route can be chosen to minimize the number and lengths of free spans [71]. When laying pipelines on the sea floor with complicated and changeable terrain, the unevenness of the sea floor must be analyzed in advance [72]. Then, the uneven routing areas should be preprocessed, which can greatly reduce the design cost and shorten the pipeline laying period.

### 2.2.2. Free-Spanning of Sea Pipe Caused by Residual or Thermal Stress

Submarine pipelines may cause residual stress in the pipeline during production, processing, transportation, and installation [73]. Residual stress in oil and gas pipelines, due to the redistribution of stress and strain and thermal expansion, causes pipeline movement. When the movement is blocked at a certain position, it may cause the pipe to locally buckle and warp, releasing stress. As a result, the pipeline is partially spanned, as shown in Figure 3.

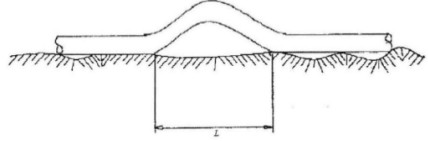

**Figure 3.** Buckling phenomenon caused by residual stress in a seabed pipeline.

Generally, avoiding pipeline span caused by residual stress or thermal stress is difficult, and the precautionary measures are complicated.

### 2.2.3. Free-Spanning of Sea Pipe Caused by Human Activities

Human activity is another one of the reasons for the span of submarine pipelines. Channel excavation, sand excavation, land reclamation, and bridge pier construction near the submarine pipeline considerably influence the hydrological conditions around the pipeline, which destroys the balance of sediment erosion and deposition. To some extent, the natural law of the original seabed or riverbed is changed, resulting in the free span of the pipelines. In the vicinity of human activities, the anchorage of ships, trawling of fishing boats, and household garbage may pose a threat to the safe operation of submarine pipelines.

## 3. Treatment Methods of Submarine Pipeline Free-Spanning

After the laying or operation of the submarine pipeline, if spanning occurs due to the erosion of the current, residual stress, or thermal stress, and the span exceeds the allowable length. The span of the pipeline then causes fatigue damage due to vortex-induced vibration under the action of the ocean current [74,75]. In addition, the span pipeline loses its landfill. If it is impacted by falling objects or anchors, it is more vulnerable to damage. The spanned pipeline is directly exposed on the sea floor surface, which can interfere with the trawl of fishing boats, tow, and even to be displaced or buckled under towing. To ensure the safety of pipeline operations and prevent oil and gas leaks, appropriate methods should be adopted to mitigate and manage the free span of the pipeline. At present, the main methods for free span treatment of submarine pipelines include: re-ditching and burying methods, structural support method, rock throwing, cement briquette, and covering with artificial grass.

### 3.1. Re-Digging and Burying

The method of re-ditching and burying involves re-ditching and filling the span of the pipeline using subsea-trenching equipment. The research on subsea trenching machinery began in the late 1940s. Since the first jet skid was used in the submarine oil and gas pipeline in the Gulf of Mexico in North America in 1946, various types of trenching equipment have been developed [76]. According to the different principles, re-digging ditches can be divided into the jet ditching method and mechanical ditching method.

### 3.1.1. Jet Ditching Method

The jet trencher sprays the soil at the bottom of the pipe with high-pressure water to form a trench under the pipe. When the suspended length of the pipeline exceeds the critical value, it will sink to the bottom of the ditch under its own gravity. Water flow is then used to back silt the pipeline to create landfill [77]. At first, the exploitation of marine oil and gas resources was mainly concentrated in offshore areas. The soil in these areas is dominated by soft silt and the working water depth is shallow. Therefore, jet trenchers were mainly used at first. The first generation jet trencher has a simple mechanical structure, as shown in Figure 4.

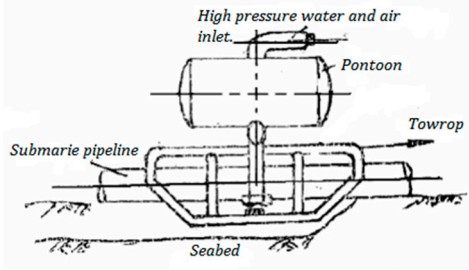

**Figure 4.** First generation jet trencher.

The second-generation jet trenchers have slide shoes added to the original structure and the shape and position of the nozzle were changed to achieve a better trenching effect. Its structure is shown in Figure 5 [78].

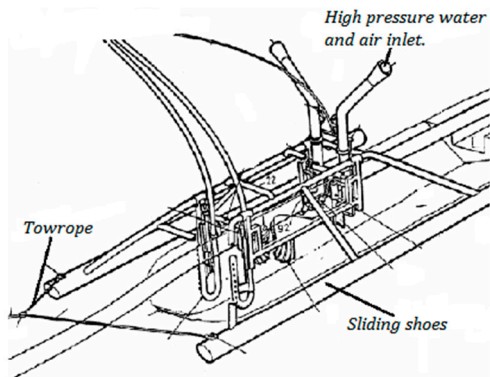

**Figure 5.** Second-generation jet trencher.

The power required for the second-generation trenchers to dig and jet trenches is still provided by marine auxiliary vessels. With the advancement of science and technology, water pumps that can be used in seawater were installed on jet trenchers, creating the modern jet trenchers, as shown in Figure 6 [79]. By installing the pump on the jet trencher, the hose that used to carry compressed water and gases can be removed, and the loss on the hose can be reduced. Although the working efficiency of the modern jet trencher has been significantly improved, it still has strict requirements for soil quality and trenching is inefficient in hard clay.

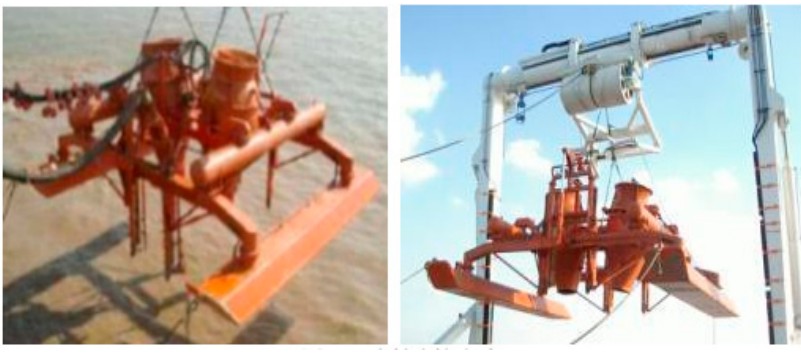

**Figure 6.** Modern jet trencher.

### 3.1.2. Mechanical Ditching Method

To enhance the adaptability of trench diggers to seabed geology, various mechanical trench diggers have been developed [80,81]. When a plough trencher is used for pipeline laying, the trencher is pulled along the pipeline-laying path by a water towboat with a tow rope, the plough is trenched on the sea floor, and the soil is turned over on the bottom of the trench. Then, the pipe is placed in the trench. Relying on the current, the mud accumulated on both sides of the ditch is flushed to bury the pipeline [82–85]. The idea for a submarine trench excavator was introduced in the 1970s. After more than 40 years of development, various types of submarine trench excavators have emerged for the optimization of seabed soil hardness, seabed working depth, and economic benefits. In 1976, the world's first pre-trenching submarine trencher was developed, which successfully dug trenches at a depth of 150 m [86]. In 1989 and 1990, Soil Machine Dynamics Ltd. (SMD) developed the modular plough system (MPS) [87,88] and advanced pipeline plough (APP) [89,90] for Stolt Offshore, respectively, as shown in Figure 7a,b. In 2004, CTC Marine developed a trench digger called advanced

multi-pass plough (AMP) [91,92]. As shown in Figure 7c, this trench digger has the addition of a high-pressure injection system similar to that of a jet trench digger, which can reduce the cutting force and cope with viscous sands via high-pressure injection on the cutting surface of the plough body. In 2006, SMD developed the latest generation of trencher, the variable multi-pass plough (VMP), as shown in Figure 7d. This trencher adopts hi-tow point technology and has variable stroke [93].

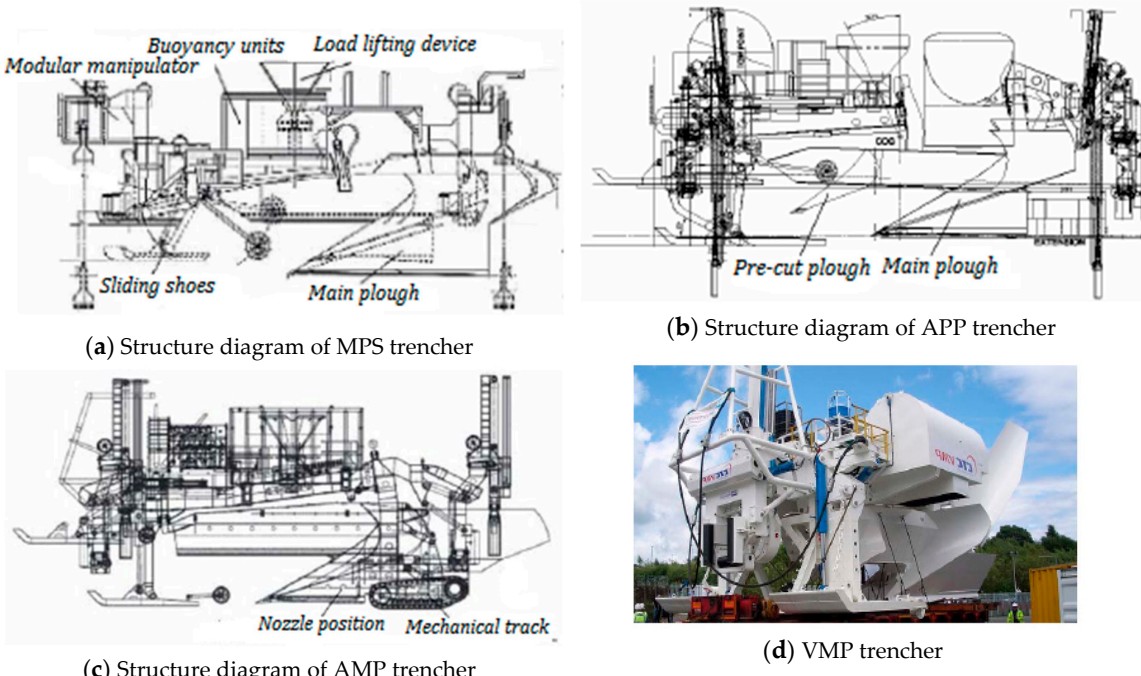

(**a**) Structure diagram of MPS trencher

(**b**) Structure diagram of APP trencher

(**c**) Structure diagram of AMP trencher

(**d**) VMP trencher

**Figure 7.** Mechanical trenchers of the modular plough system. Abbreviations: MPS, modular plough system; APP, advanced pipeline plough; AMP, advanced multi-pass plough; VMP, variable multi-pass plough. (**a**) Structure diagram of MPS trencher, (**b**) Structure diagram of APP trencher, (**c**) Structure diagram of AMP trencher, (**d**) VMP trencher.

### 3.2. Structural Support Methods

The structural support method is implemented by installing a support structure below a span pipeline; thereby, reducing the span length and ensuring the safety of the spanning pipeline. At present, the main structural support methods include grouting bag support, underwater pile support, and throwing stone support.

### 3.2.1. Grouting Bag Support Method

The grouting bag method involves placing a supporting shell membrane on the surface of the sea floor and then injecting mud into the grouting bag using the grouting equipment on the mother ship to complete the support of the span section of the pipeline [94]. The advantages of this method include simple and easy construction, high economic benefits, and no damage to the pipeline. The disadvantage is that, over time, the scouring action of waves and currents may cause the grouting bag to sink. After sinking, it no longer provides support for the spanned pipeline. Found Ocean, a British company, is the world's largest offshore construction and grouting company that provides design, manufacturing, and installation services for various types of grouting bags. Figure 8a shows Found Ocean's J-type grouting bag, which was successfully assembled with remote operated vehicle (ROV) at a depth of 1244 m underwater. Figure 8b depicts a schematic diagram of a grouting bag laying device invented by Wang et al. [95] in 2015, and Figure 8c depicts the grouting bag laying device invented by Song et al. [96] in 2016.

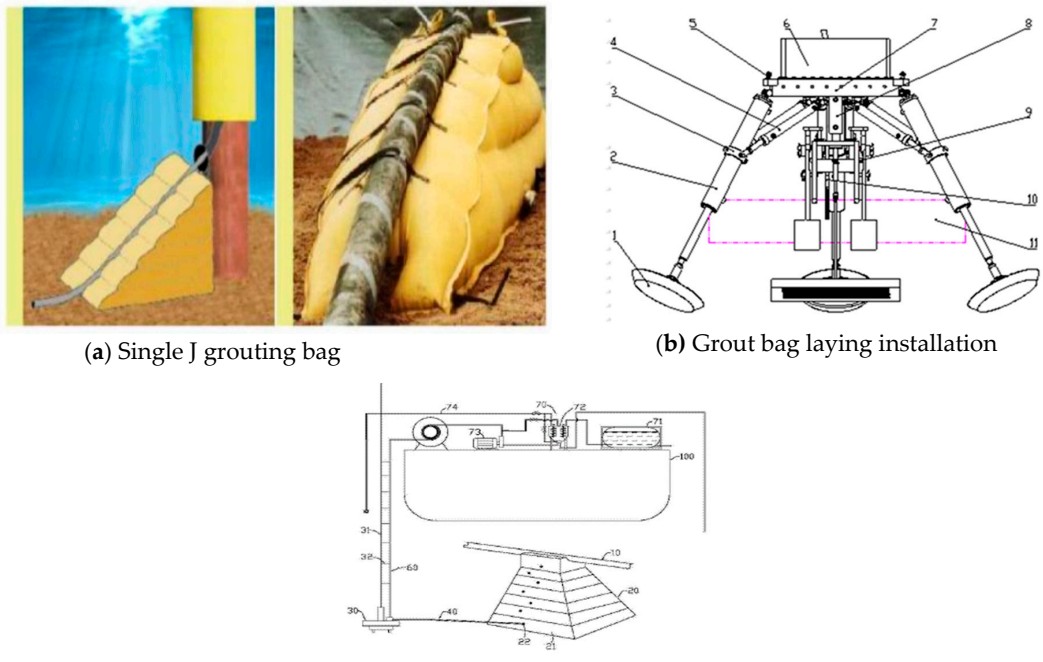

(**a**) Single J grouting bag

(**b**) Grout bag laying installation

(**c**) Grout bag-laying mechanism

**Figure 8.** Installation services for various types of grouting bags. (**a**) Single J grouting bag, (**b**) Grout bag laying installation, (**c**) Grout bag-laying mechanism.

### 3.2.2. Underwater Pile Support Method

The underwater pile support method involves alternately setting underwater support steel pipe piles on both sides of a pipe span. To reduce lateral and longitudinal vibration amplitudes, the setting distance is determined by the maximum allowable span length of the pipe and the strength of the pipe. Both the Gulf of Mexico project at the Canyou Express oilfield project and the West Africa Ceiba Field project used submerged steel piles to treat spanning pipelines. This method has the advantage of higher reliability. However, if multiple steel pipe pile supports are needed, this method is time-consuming and expensive. Figure 9 depicts a supporting pile structure that is currently common [97–99].

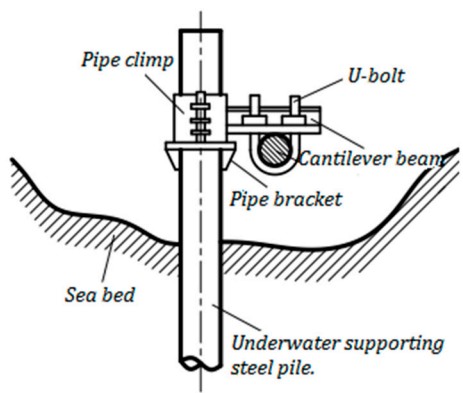

**Figure 9.** Underwater support piles.

The current supporting piles are mostly cantilever beam structures. To fix the cantilever beam, a pipe bracket is generally pre-welded on the supporting pile. After the support pile is driven, the diver measures the relative position of the tube holder and the suspended pipe. Then, the prefabricated cantilever beam and the pipe clamp are fixed on the pipe bracket according to the actual measurement size. Finally, the diver fixes the suspended pipeline to the cantilever beam with high-strength U-shaped

bolts. This method is mainly manual and difficult for divers. For the treatment of deep-water span pipelines, the operation by divers is more difficult.

### 3.2.3. Riprap Support Method

Stone throwing is one of the most traditional free-span governance measures for subsea pipelines. It has been widely used in the development of offshore oil and gas resources around the world [100–102] and has broad application prospects. Under extreme hydrodynamic loading conditions, stabilizing large diameter subsea natural gas pipelines has proven to be a challenge in northwestern Australia. The tropical storms that affect the area between November and April each year can produce wave heights of more than 30 m and storm currents of 2 m/s or more. Therefore, at shallow water depths, usually below 40–60 m, submarine pipelines may be subject to very high hydrodynamic loads, which may cause significant lateral movement. To reduce the risk of mechanical damage to the pipeline due to excessive lateral movement, mined and graded rock is usually poured onto the pipeline as a secondary stabilization solution.

In the Ormen Lange project in Norway, the rubble method was adopted to govern the free-span of a submarine pipe [103]. The stone throw method fills the scour pit by throwing crushed stones into the span section of the pipeline, thus supporting the pipeline and reducing the length of the pipeline span. The method is simple in construction and does not require engineering design. During construction, only the concrete position of the riprap must be determined according to the investigation results. For shallow water pipeline span, the diver can directly complete riprap operations. However, a riprap guide device must be used in deep water, otherwise the debris may drift away from the designated drop site due to the action of waves and currents. When the water depth exceeds 100 m, due to the large riprapping diversion device, a crane with a certain lifting capacity is required for onsite installation [104].

### 3.3. Covering Bionic Water Plant Method

Bionic aquatic plant technology on the seabed is a kind of anti-scour measure developed on the basis of marine bionics [105]. The bionic anti-scour system includes bionic seaweed, bionic seaweed installation base pad, a specifically designed submarine anchorage device, and other parts, as shown in Figure 10.

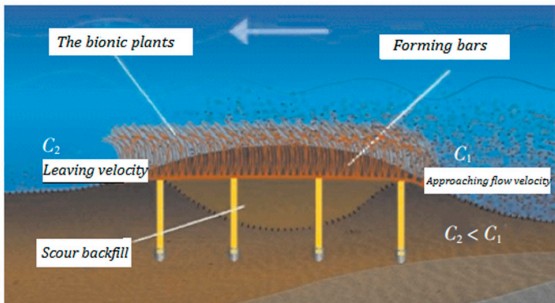

**Figure 10.** Bionic aquatic plant action mechanism.

After the bionic grass and its mounting mat are firmly anchored at a predetermined position on the sea floor where erosion needs to be prevented or controlled, the bottom water flows over this bionic water grass. Due to the flexible viscous damping effect of the bionic grass, the flow velocity is reduced, which slows down the scouring effect of the water on the sea floor. Due to the decrease in the flow velocity and the obstruction created by the bionic aquatic plants, the sediment entrained in the water stream is continuously deposited on the bionic aquatic plant installation mat under the action of gravity, gradually forming seabed sandbars enhanced by bionic grass. As such, the scour of the sea floor near the submarine pipeline is suppressed [106]. When using bionic waterweed technology to treat

submarine overhang pipelines, the scoured trenches at the bottom of the pipeline need to be backfilled first with sand and gravel. After leveling, the bionic waterweed is installed on the backfill layer. Bionic water grass technology has achieved good results in the management of submarine spanning pipelines in Shengli Chengdao Oilfield and Nanpu Oilfield [107]. Li et al. [108,109] invented a new floating curtain-descaling device that can slow down the flow rate of water. When the water-carrying sediment flows through the sand window, the flow rate decreases and the sediment can be deposited under the action of gravity, thus covering the pipe.

### 3.4. Self-Embedding Choke Plate Method

Due to the high cost of trench-digging equipment, the self-burial of sea pipes has been studied. Pipes laid on the sandy seabed gradually sink due to erosion after a period of time. Since the 1980s, experimental studies and theoretical analyses on the sinking and self-burying of sea pipes have been conducted. Through numerical simulation, Cheng and Chew concluded that the choke plate significantly increases the pressure on the upstream surface of the pipeline and reduces the pressure on the back basement of the pipeline [110]. Hulsbergen et al. studied the scour effect of submarine pipelines with the addition of baffle, and concluded that adding baffle above submarine pipelines can increase the scour depth and width of ocean currents on the seabed. Hulsbergen et al. [111,112] and Gokce et al. [113] studied the impact of choke plate on pipeline scour under unidirectional flow and wave action, respectively, and found that the depth of pipeline scour increased with additional choke plates. Yang [114] studied the scour of the seabed after installing rigid and flexible baffle plates under the action of waves and current through physical experiments. Through physical model tests, Han et al. [115] compared the pressure distribution and scour pit around submarine pipelines with flexible choke plates, rigid choke plates, and open flow plates installed under the action of unidirectional constant flow. Zhang et al. [116] examined the influence of choke plates on submarine pipelines under a wave environment by combining numerical simulation and experiments, and proposed that the increase in choke plate height could increase the maximum horizontal wave force, while the maximum vertical wave force gradually decreased. Chiew [117,118] studied the impact of the installation orientation of the choke plate on the scour of submarine pipelines under unidirectional flow. Jiang et al. [119] simulated and analyzed submarine pipelines arranged at angles of 30°, 60°, and 90° for open-flow plate submarine pipelines, baffle plates, and seawater flow direction, and found that an baffle installation angle of 60° is the most conducive to the sinking and self-burying of pipelines. The choke plate was designed by a Dutch company (SPS) and developed in the United States. The first field test was conducted in the North Sea in 1998. A standard choke plate consists of a fin and a template. The structure of the choke plate is shown in Figure 11. The vertical fin can significantly change the distribution of the flow field around the pipe; thus, accelerating the self-burying process of the pipe. In 2003, China cooperated with an American choke plate company as well as a Dutch company (Advanced Consultancy Romke Bijker, ACRB), to apply choke plate technology in the Hangzhou Bay pipeline project [120]. This technology has been proven to have good economic benefits and produce an environmental protection effect.

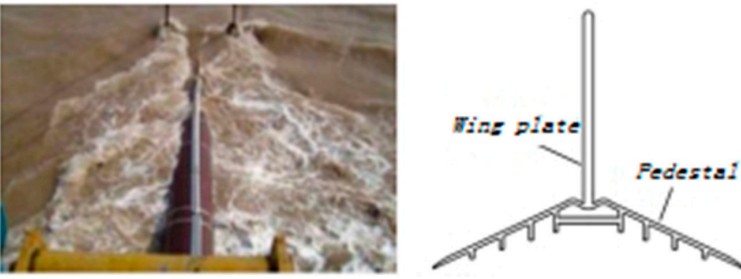

**Figure 11.** Installation and structure of a choke plate.

### 3.5. Self-Buried Subsea Pipes Caused by Seabed Liquefaction

In engineering practice, the liquefaction of soils under dynamic loads, such as waves and earthquakes and especially non-cohesive sand, is a widespread phenomenon [121–123]. Since liquefied soil behaves as a liquid, it does not have any carrying capacity to support any structure built upon it. Therefore, the pipeline will sink under its own gravity, which is another principle of pipeline self-burial. There are two types of liquefaction mechanisms for the sea floor: transient liquefaction and residual liquefaction. The transient liquefaction of the subsea foundation is mainly related to the phase lag between the dynamic pore pressure on the ocean floor and the dynamic pressure caused by the propagation of the ocean floor waves. Residual liquefaction is mainly caused by the increase in soil pore pressure under waves or seismic load. With the increase in soil pore pressure, the effective stress of contact between soil particles gradually decreases. When the contact effective stress is zero, the soil liquefies.

Based on engineering experience, some scholars proposed methods to evaluate soil liquefaction sensitivity using soil property parameters [124–127]. The experience-based prediction method for soil liquefaction sensitivity is only a qualitative analysis, only indicating the possibility of soil liquefaction. Regarding the study of quantitative soil liquefaction criteria, Okusa [128] first proposed a one-dimensional liquefaction criterion based on vertical effective stress. Based on this, Tsai [129] extended it to three-dimensional situations. Yamazaki [130], and others, proposed a liquefaction standard based on dynamic pore pressure, which Jeng [131] extended to the three-dimensional case. Nataraja [132] analyzed the failure cases of the seabed foundation, improved the liquefaction analysis method caused by earthquakes, and analyzed the relationship between wave-induced seabed liquefaction and wave period, water depth, and seabed strength. Subsea liquefaction cannot only be found in buried pipes, but also an important cause of instability of marine structures such as buried pipelines [133].

### 3.6. Other Governance Methods

Many treatment methods are available for submarine spanning pipelines. In addition to the above several methods, engineers and technicians have formulated many new treatment schemes for different types of spanning, different seabed topographies, and different seabed soil properties, such as the blowing mud settlement method, the flexible hose bond method, the cement compaction method, and so on.

Liu [134] introduced a new method of mud blowing sedimentation for sea pipes by analyzing the suspended section of a submarine pipeline in the Ledong gas field. By using a high-pressure water gun to scour the sediment in the mud section of the pipe on the top of the "hill", the suspended section of the pipe gradually settles. Finally, the free-spanning length of the pipeline is reduced and the free-spanning section of the pipeline is treated effectively. This method is suitable for seabed surfaces with soft sediment. Lin et al. [135] described a method for processing the suspended span of a sea pipe based on jet flow. They introduced the steps to apply this method to the treatment of three types of suspended span sea pipe: L-type, U-type, and W-type. The flexible hose bond method is suitable for the bond section of offshore platforms and submarine horizontal pipelines, as well as the situations where the seabed topography fluctuates markedly. When this treatment plan is implemented, the suspended section of the pipeline must be removed, and the hose must be redesigned and installed according to the current seabed status after scouring. Due to the flexible nature of the hose, it can change with the change in the terrain during lying, and has good anti-fatigue failure characteristics [136]. However, the pipeline must be shut down during the implementation of the plan. When the free-spanning height is very low, about a few centimeters, heavy ballasts can be used to cover the pipeline to avoid the occurrence of vortex-induced vibration. When the height of the suspended pipe is large, the overlaying of heavy ballast on the suspended pipe is likely to cause bending deformation of the pipe [137], so this method is no longer used.

In shallow waters, the sea floor is active, involving sand, strong currents, and moderate wave activity. Extensive sand wave fields have been found in the southern North Sea, the South China Sea, and the Persian Gulf. The migration of sand waves causes buried pipelines to be exposed. The burial depth below the sand wave groove envelope can increase the risk of pipeline exposure and free development. Therefore, a suitable trenching depth can be selected to reduce the risk of pipeline exposure. However, due to cost considerations, limited dredging of intersecting sand wave peaks in critical areas is often used in engineering to perform system corrections, or wait-and-see methods are adopted if direct danger is not anticipated. Intervening quickly when problems arise can provide huge benefits. This method requires a reliable assessment of the evolution of the bed and the dynamic load that the pipeline should bear [138].

### 3.7. Comparative Analysis of Various Governance Methods

In the governance of free-span pipelines, we facilitate the selection of various methods by engineering and technical personnel by comparing the advantages and disadvantages of several governance schemes commonly used in engineering practice in Table 2.

**Table 2.** Comparison of advantages and disadvantages of each scheme.

| Management Plan | Advantages | Disadvantages |
|---|---|---|
| Ditching plow method | Applicable to seabeds of various soil types, with low requirements on operating sea conditions | Required traction is large, the pipeline crossing other sea pipe or cable cannot be used, the cost is high |
| Jet ditching method | Suitable for soft seabed, small traction required | Efficiency of trenching on hard seabed is low; when water depth is over 100 m, wave current and sea current actions on water pipe or air hose may cause instability of trencher foundation |
| Grouting bag support method | Construction is simple, management cost is low, the implementation does not need the pipeline to stop production | Long-term erosion of the wave current may cause the grouting bag to sink and lose support to the pipe |
| Underwater pile bracing | Can be adapted to undulating seabed terrain and handle high free-spanning pipelines | High cost, small protection range, support pile, as a new structure, may cause secondary erosion |
| Ripped-rock method | Simple construction, easy access to raw materials, and can handle a wide range of free-spanning | When the water is deeper, a riprap guiding device should be used |
| Bionic water plant method | Prevents pipeline erosion, promotes sedimentation, good treatment effect, no secondary erosion | High cost, time consuming, and cannot be used as a free-spanning emergency treatment measure |
| Spoiler self-buried method | Easy installation, no need for offshore operation | Not suitable for rock and clay seabed |
| Flexible hose bonding method | Flexible, can change with the seabed topography and effectively resist fatigue damage | The hose should be redesigned according to the scoured terrain, and the pipeline must be shut down during implementation |
| Heavy ballast method | Simple construction, high economic benefits | Only applicable to the free-spanning height of a few centimeters or so |

At present, many methods are available for managing free spanning submarine pipelines. For different marine environments, different protection measures need to be adopted to ensure the safe operation of pipelines. Sometimes, to achieve the best governance results, multiple governance methods need to be combined. We provide the following governance recommendations:

(1) For submarine pipelines with large free-span height and long free-span distance, adding support using the staggered arrangement of underwater piles can be considered first. If the free span is caused by the climbing of the pipeline and the soil on the hillside is soft, the height of the free

span can first be reduced using blown mud to subsidize the sea pipe, and then treating it with the structural support method.

(2)     When laying pipelines on the seabed with large water flow speed and soft soil, bionic waterweed cover can be used to prevent pipeline erosion, or a baffle plate can be installed on the pipeline to promote the pipeline to self-bury.

(3)     For the free span of long length but small diameter pipelines, the trenches can be re-ditched or the ballasting method can be used for heavy objects;

(4)     For short-distance, free crossing pipelines in shallow waters, throw-filled crushed stone or grouting bag support can be used reasonably.

(5)     For the transition between the offshore platform riser and the submarine horizontal pipeline, a flexible hose crossover can be used.

(6)     For pipelines near offshore platforms, sandbags should be used to backfill all the spans of the pipeline, and then the pipelines should be covered with cement briquettes to improve the ability of the cover to resist erosion and external impact.

In the actual treatment of free spanning pipelines, we need to comprehensively consider factors such as the depth of the seawater, the nature of the subsea soil, the cost of treatment, and the effect of the treatment to select the best treatment solution.

## 4. Conclusions

The purpose of this study was to summarize and analyze the causes of submarine suspended span pipelines and the available treatment measures. Submarine pipelines are indispensable in the exploitation of offshore oil and gas resources. Since the 1950s, numerous reports have been published about submarine pipeline damage caused by scouring and free-spanning. According to Minerals Management Set (MMS) statistics of submarine pipeline failure accidents in the Gulf of Mexico from 1967 to 1987, a total of 690 submarine pipeline failure accidents occurred in 20 years, with an average annual failure accident of 35 cases [139]. Wang [140] investigated 61 submarine pipelines in China's Chengdao Oilfield and found that only five pipelines were not scour and suspended, accounting for only 8%. These data illustrate the need for in-depth research on the causes and treatment measures for the free span of subsea pipelines.

The reasons for the free span of submarine pipelines include the scour of wave flow, the fluctuation of seabed topography, residual stress, or thermal stress of pipelines, and human activities, among which the scour caused by waves and currents is the most important. The study of scour initiation and maximum depth is crucial for the prevention and treatment of scour of submarine pipelines. Due to the many and complex factors affecting scour, the current research is limited by the research means, observation, and measuring instruments. We are not yet able to fully consider all the influencing factors, so only reasonable simplified methods can be adopted for analysis and prediction.

At present, the submarine suspended span pipeline treatment methods mainly include the re-digging trench burying method, structural support method, bionic water plant method, choke plate self-burying method, flexible hose bridging method, and heavy ballast method. Although many experts and scholars have formulated and implemented many pipeline free-span management methods, the most effective protection measures have not yet been found. Current treatment methods generally have disadvantages such as high cost and a large amount of underwater operation by divers. With the exploration of oil and gas resources in deeper waters, management measures with a high degree of automation and good economic benefits are required.

**Author Contributions:** Conceptualization, B.Z.; methodology, R.G.; formal analysis, T.W.; writing—original draft preparation, Z.W. All authors have read and agreed to the published version of the manuscript.

**Funding:** This research was funded by National Natural Science Foundation of China (NNSFC), grant number: 51879063 and 51479043. The views expressed here are those of the authors alone.

**Conflicts of Interest:** The authors declare no conflict of interest.

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
