# Peer review of "Causes and Treatment Measures of Submarine Pipeline Free-Spanning"

_jmse, doi:10.3390/jmse8050329_

Round 1

Reviewer 1 Report

see attached report

Author Response

I am very grateful to your comments for the manuscript. According you’re your advice, we amended the relevant part in manuscript. Some of your questions were answered below.

To Reviewer:

Main issues
Opinion 1: Completeness of context analysis. As clarified in my first-round review, if this wants to be a true review paper the topics of interest should be discussed in view of the global literature rather than just “home” (i.e. Chinese) literature (see also “Specific Issues”)..

Responses to 1: According to the suggestion, the revised article added the relevant theories of clean water scour and live bed scour in lines 151-1162 and added the research results of Dey ; Moncada-M et al. Lines 200-204 supplement the methods for erosion prediction under the pipeline, including methods of experimental research and numerical calculation, and methods of artificial intelligence, and provide references. On lines 457-480, the author rewrites the study of seabed erosion, while deleting Chinese literature and supplementing global literature in this area.

Opinion 2: The Presentation As clarified in my first-round review, too generic sentences should be avoided, as useless, in favor of focussed analyses of the fundamental mechanisms at play.

Responses to 2: Lines 506-512; lines 524-535 of the original text, and sentences with little meaning have been deleted.

Opinion 3: Use of the English language requires some significant improvement. The assistance of a professional translator is now essential.

Responses to 3: According to suggestions, the use of English will seek the assistance of professional translators.

Specific issues
Opinion 1: line 109. Perhaps, it is meant “pipeline axis” here? Please, check and amend;

Responses to 1: After inspection, the pipeline axis is more accurate here, and it is modified on line 111 of the revised article.

Opinion 2: lines 147-151. The separation between “clean water” and “live bed” (not “moving bed”) scour has been introduced and discussed much earlier and in much more fun- damental works than that here mentioned. Please, amend wording and referencing;

Responses to 2: Lines 151-1162 of the revised article provide a more complete description of clean water scour and live bed scour, and have changed inappropriate words.

Opinion 3:lines 153-154. What is the “scour problem under the action of water”? Was it mentioned to be “scour problem under the action of currents”?

Responses to 3: Upon inspection, this should be "scour problem under the action of currents", at line 165 of the revised article.

Opinion 4:lines 156-160. “wave water particles” has no meaning and should be replaced by “water particles”;

Responses to 4: Lines 166-168 of the revised article, replacing "wave water particles" with "water particles".

Opinion 5:lines 289-295. Description of these specific devices and their technical details is of little use, as many similar ones are available around the world. As already men- tioned, a review paper must be as general as possible, rather than site (China) specific;

Responses to 5: Deleted lines 289-295 of the original text.

************************************************

We would like to express our great appreciation to you and reviewers for comments on our paper. Looking forward to hearing from you.

Thank you and best regards.

Yours sincerely,

Zhuo Wang

Eail: wangzhuo_heu@hrbeu.edu.cn

Apr.03.2020

Reviewer 2 Report

The work reviews the causes of submarine pipeline free-spanning and some methods used to address this problem. Generally speaking, the paper is well organized. It offers a suitable overview with adequate choice of references. It would be of interest to the researchers in this field of work. However, prior to publication, certain aspects must be addressed:

1. The quality of English language. There are some very basic errors and even sentences that are grammatically not sentences. For instance, already in the abstract, the reader may find: "with a convenient to 21 understand" (convenient is an adjective), or "to choice the suitable" (choice is a noun, not a verb). An example of a sentence, that is actually not a sentence: "For submarine pipelines with significant suspension." (page 2, line 47). Those were only examples, and there are number of those in the paper. 

2. Page 2, line 55: The authors use "at home". I assume they mean "in China". If yes, this is how it is supposed to be written, because there is no need for reader to guess or conclude what "at home" means. 

3. The authors use "suspension span" and "free-spanning". It would be good to go for one term and use it consistently. Also, it would be good to first introduce the term and then use it. 

4. As a rule, all terms used in an equation should be either introduced prior to the equation or immediately after it. This is not the case with Equation 1 and 2. And even worst, some of the terms are not explained at all. When explaining quantities, do not start the sentence with "Where", because it has then a form of a question. You may use "where" as a continuation of the sentence that contains the equation. 

5. Some of the figures appear to be scans of rather low quality. If they are scans, maybe the source should be referred to? Another aspect is that the quality of the figures could and should be better.

6. Page 4, line 141 - he authors refer to "consistency index of cohesive soil, but never introduced the term. Similarly, several lines later the authors use Θ, before introducing it, but then introduce it by a sentence completely put in parenthesis, which is not appropriate. 

7. The rows in Table 1 should be in the same line. Check, for instance, rows 1 and 2. 

8. Regarding the methods to resolve the issue of free-spanning of submarine pipelines, I am missing in this paper a short analysis/discussion by the authors (could be in conclusions, as a separate paragraph) regarding the advantages and disadvantages of the methods, suggestions for the choice and similar. This is exactly the aspect that would introduce new quality into the overview provided. 

Author Response

Responses to the reviewer 2

I am very grateful to your comments for the manuscript. According you’re your advice, we amended the relevant part in manuscript. Some of your questions were answered below.

Opinion 1: The quality of English language. There are some very basic errors and even sentences that are grammatically not sentences. For instance, already in the abstract, the reader may find: "with a convenient to 21 understand" (convenient is an adjective), or "to choice the suitable" (choice is a noun, not a verb). An example of a sentence, that is actually not a sentence: "For submarine pipelines with significant suspension." (page 2, line 47). Those were only examples, and there are number of those in the paper. 

Responses to 1: According to the recommendations, the use of English has been appropriately modified, and the use of English will seek the assistance of professional translators.

Opinion 2:Page 2, line 55: The authors use "at home". I assume they mean "in China". If yes, this is how it is supposed to be written, because there is no need for reader to guess or conclude what "at home" means. 

Responses to 2: The sentence on line 55 was modified, on line 55 of the revised article.

Opinion 3:The authors use "suspension span" and "free-spanning". It would be good to go for one term and use it consistently. Also, it would be good to first introduce the term and then use it. 

Responses to 3:Lines 57-58 of the revised article provide a brief explanation of "free-spanning". And by searching, I changed the "suspension span" elsewhere in the article to "free-spanning".

Opinion 4: As a rule, all terms used in an equation should be either introduced prior to the equation or immediately after it. This is not the case with Equation 1 and 2. And even worst, some of the terms are not explained at all. When explaining quantities, do not start the sentence with "Where", because it has then a form of a question. You may use "where" as a continuation of the sentence that contains the equation. 

Responses to 4: Lines 106-113 of the revised article explain the remaining terms in Equation 1 and 2, and explain the quantities with the beginning of “Here”.

Opinion 5:Some of the figures appear to be scans of rather low quality. If they are scans, maybe the source should be referred to? Another aspect is that the quality of the figures could and should be better.

Responses to :5:I have appropriately deleted the numbers in the article that are difficult to find, such as deleting the text at line 510 of the original text.

Opinion 6:Page 4, line 141 - he authors refer to "consistency index of cohesive soil, but never introduced the term. Similarly, several lines later the authors use Θ, before introducing it, but then introduce it by a sentence completely put in parenthesis, which is not appropriate. 

Responses to 6:By re-reading the article from Kumar et al., The quotes were newly described in lines 142-144 of the revised article, and the brackets in the sentence explaining the parameters were removed in lines 150-151.

Opinion 7:The rows in Table 1 should be in the same line. Check, for instance, rows 1 and 2. 

Responses to 7:The format of Table 1 was checked and modified.

Opinion 8: Regarding the methods to resolve the issue of free-spanning of submarine pipelines, I am missing in this paper a short analysis/discussion by the authors (could be in conclusions, as a separate paragraph) regarding the advantages and disadvantages of the methods, suggestions for the choice and similar. This is exactly the aspect that would introduce new quality into the overview provided. 

Responses to 8:According to the recommendations, in the revised article at lines 512-535, six new recommendations for free spanning pipeline governance were added.

************************************************

We would like to express our great appreciation to you and reviewers for comments on our paper. Looking forward to hearing from you.

Thank you and best regards.

Yours sincerely,

Zhuo Wang

Eail: wangzhuo_heu@hrbeu.edu.cn

Apr.03.2020

Round 2

Reviewer 1 Report

The paper has been further improved upon revision. However, significant improvements in the content exposition and use of the English language are still required.

Reviewer 2 Report

The authors have improved the paper significantly. But they still need to improve the English language and to write the paper more carefully. For instance, after many sentences there is no blank (empty space) after the full stop. Or, there are places such as these:

"Through reading a lot of relevant literatures; it’s found that..." (inappropriate use of ; and also "literatures"),

"Mattioli et al.[49-50] Used particle..." (capital U)

"Sumer B. et al.[67] Reviewed..." (capital R)

etc. 

In the previous review, I noticed the use of "at home" by the authors. They have corrected it at that specific place I mentioned, but left it at other places. So, you may still find in the paper "at home and abroad" and also "domestic". 

When explaining the terms in equations (1) and (2), the authors should write them in exactly the same manner as in the equations - for instance, ΔCp is written italic in the equations, but not in the text. 

"s is the relative specific gravity of the sand and water" - I don't understand this. Do you mean the relative specific gravity of the sand with respect to water? 

There is no reason to explain a term two times (for instance D). 

I didn't understand the reply by the authors related to the quality of the figures and the possible need to refer to the source in case the figures are scanned from some other sources (as it seems to be). But if the journal is fine with that, it is ok. 

The authors should try to avoid mixing passive and active voice, such as here: "For submarine pipelines with large free span height and long free span distance, the support method of staggered arrangement of underwater piles can be considered first (passive!); If the free span is caused by the climbing of the pipeline and the soil on the hillside is soft, you can first reduce (active!) the height of the free..."

"In the actual treatment of free spanning pipelines, we need to comprehensively consider factors such as the depth of the seawater, the nature of the subsea soil, the cost of treatment, and the effect of the treatment in order to select the best treatment solution." This is the last paragraph prior to conclusions, but almost the same text repeats in conclusions, in the middle of the last paragraph!

The authors should correct these details and then the paper can be accepted for publishing without further review.  

This manuscript is a resubmission of an earlier submission. The following is a list of the peer review reports and author responses from that submission.

Round 1

Reviewer 1 Report

see attached report
